# Comparative Effectiveness of an Autologous Dentin Matrix for Alveolar Ridge Preservation

**DOI:** 10.3390/medicina60081280

**Published:** 2024-08-08

**Authors:** Nikolai Redko, Alexey Drobyshev, Thanh Hieu Le, Dmitry Lezhnev, Roman Deev, Ilya Bozo, Andrey Miterev, Sergey Shamrin, Yaroslav Skakunov, Davronbek Meliev

**Affiliations:** 1Department of Maxillofacial and Plastic Surgery, Federal State Budgetary Educational Institution of Higher Education “Russian University of Medicine” of the Ministry of Healthcare of the Russian Federation, 127006 Moscow, Russia; dr.drobyshev@gmail.com (A.D.); dr.miterev@mail.ru (A.M.); new06f1@mail.ru (S.S.); dr.skakukov@mail.ru (Y.S.); dr.meliev@mail.ru (D.M.); 2Department of Pathological Anatomy, North-Western State Medical University Named after I.I. Mechnikov, 191015 Saint Petersburg, Russia; 3Department of Reconstructive and Plastic Surgery, Petrovsky National Research Center of Surgery, 119435 Moscow, Russia

**Keywords:** alveolar bone loss, alveolar ridge augmentation, bone regeneration, biocompatible materials, tooth extraction

## Abstract

An urgent issue is the preservation or reconstruction of the volume of bone tissue in planning and surgical treatment in the fields of medicine, such as traumatology, orthopedics, maxillofacial surgery and dentistry. After tooth extraction, resorption of the bone tissue of the alveolar crest of the jaws occurs, which must either be further eliminated by performing additional operations or using osteoplastic material for socket preservation at the extraction stage. *Background and Objectives:* The aim of the study was a comparative analysis of various osteoplastic materials used to preserve the volume of bone tissue in the preimplantation period. *Materials and Methods:* As part of the study, 80 patients were treated, who underwent socket preservation using xenografts, plasma enriched with growth factors, an autologous dentin matrix (ADM) and hydroxyapatite. *Results:* The results of the treatment 16 weeks after removal were comprehensively analyzed using a morphometric analysis of the bone’s volume, cone beam tomography and morphological examination of burr biopsy specimens, as well as by determining the stability of the installed implant at different stages of treatment. *Conclusions:* The lowest level of bone tissue resorption according to the CBCT data was noted in the ADM and xenograft groups. It should be noted that the use of osteoplastic material in jaw surgery when reconstructing alveolar defects is an essential procedure for preventing the atrophy of bone tissue.

## 1. Introduction

A World Health Organization study estimated that at least 3.5 billion people in the world suffer from oral diseases. The main causes of tooth loss are dental caries, which affects 2.5 billion people, and periodontal disease, which affects 1 billion people [1]. Tooth loss can be a detrimental factor in quality of life, as it can lead to difficulty in eating, breathing, speech and psychological well-being [2,3,4,5]. Currently, the use of dental implants has become a key component of the full rehabilitation of patients with complete and partial edentulism [6,7]. According to the annual report of Straumann (Switzerland), about 27 million dental implants are installed annually in the world to restore chewing function in patients [8].

According to the scientific literature, the loss of any tooth can be accompanied by a pronounced resorption of the alveolar ridge in the corresponding area, which leads to the formation of various defects [9,10]. In the postextraction period, resorption of bone tissue 4 months after tooth extraction is up to 45% horizontally and 43% vertically, which forces additional surgical interventions or dental implantation in more complex clinical situations [11,12]. The goal of preserving the alveolar ridge is to minimize a reduction in its size after tooth extraction by placing bone graft material in the socket of the extracted tooth [13,14]. Currently, there is a wide range of materials used in oral surgery. These are, first of all, autologous, xenogenic, allogeneic, artificial and gene-activated bone grafts, which differ in their origin, properties and form of release [9,15,16,17,18,19].

Currently, autogenous bone grafts are the only materials considered to be the “gold standard” for reconstructive interventions in the oral cavity, and have osteoinductive, osteoconductive and osteogenic properties [9,11,20]. At the same time, a number of studies have noted that when using bone autografts, the main disadvantage is the need to collect these grafts from various donor sites of the patient, which can be accompanied by various negative consequences, as well as a high probability of subsequent resorption with a loss of up to 40% of the initial bone volume during the first 3 months after the operation [13,21,22].

Separate studies have been devoted to the use of an extracted tooth as a material for restoring the alveolar crest of the jaws in the preimplantation period. In their studies, Schwarz et al. indicated the effectiveness of using an autogenous dentin block as a graft to increase the alveolar crest of the jaws [18]. After mechanical and antiseptic processing, the extracted tooth is a safe osteoplastic material for patients [19,23]. The method of closing the defect with a fragment of an extracted tooth for the prevention of atrophy of the jawbone tissue was recommended by Neumeyer et al. [17]. Another direction in the use of an extracted tooth is the use of a crushed autologous dentin matrix (ADM) for the preservation of the alveolar process in the preimplantation period [16]. Considering its biological characteristics, lack of allergic reactions and accessibility to the clinician and patient, an autologous dentin matrix can be considered as a possible option as a graft for graft preservation in oral surgery [24].

Despite the considerable amount of research on this problem, the improvement of the preimplantation preparation of patients with adentia after tooth extraction continues to be an extremely urgent task. The solution to this problem requires a comprehensive approach based on a unified assessment of the success of dental implantation in the postextraction zone. The aim of our study was to conduct a comprehensive comparative analysis of osteoplastic materials after preservation of the extraction wells of extracted teeth. The null hypothesis was that the ADM technique could be used in patients and the newly formed bone would be sufficient for the implant placement. 

## 2. Materials and Methods

### 2.1. Study Design

The study was conducted in accordance with the rules and principles of evidence-based medicine in compliance with the requirements of the Declaration of Helsinki of the World Medical Association 2013 and was approved by the Interuniversity Committee on Ethics (protocol No. 04-18, dated 19 April 2018). The current study protocol was assessed and authorized by Moscow State University of Medicine and Dentistry (NCT06541236).

The recruitment of patients and their treatment was carried out within the framework of the University Clinic of A.I. Evdokimov Moscow State University of 67 from 2018 to 2020. The study type was a randomized clinical intervention in a prospective longitudinal study. The study included 80 patients in need of the extraction of a tooth, and further dental implantation following the proposed inclusion criteria. After selection, complex oral rehabilitation was carried out to restore chewing function. All patients (*n* = 80) were divided into 4 equal groups, depending on the preservative material used. In total, 151 teeth were extracted. In the first group, preservation was carried out using a Cerabone xenograft (Botiss, Berlin, Germany). In the second group, plasma rich in growth factors (PRGF) was used, obtained from the patient’s venous blood 20–30 min before the tooth’s extraction (BTI Endoret, Álava, Spain). The third group consisted of patients who underwent preservation of the extracted tooth socket with a crushed autologous dentin matrix (ADM) obtained from their own tooth using a Smart Dentin Grinder (KometaBio Inc., Fort Lee, NJ, USA). In the fourth group, a material based on hydroxyapatite, “Kollapan-L”, with lincomycin (Intermedapatit, Moscow, Russia) was used as a preservation graft.

Four months after the tooth’s extraction, dental implants were installed according to the standard ITI protocol [25]. In the postoperative period, a standard course of antibacterial and anti-inflammatory therapy was carried out.

### 2.2. Inclusion Criteria

To be included in the study, the treatment plan was discussed with each patient before surgery, and an informed release form approved by the ethics committee was signed. Moreover, they met the following criteria: (1) age 18–70 years; (2) the presence of indications for tooth extraction, namely periapical periodontitis, fracture of the root or crown of the tooth without the possibility of rehabilitation, or chronic periodontitis and (3) satisfactory oral hygiene.

### 2.3. Exclusion Criteria

Patients under 18 years of age, pregnant women, and patients with severe comorbidity in the stage of decompensation were not included in the study.

### 2.4. Evaluation Criteria for the Results

Before removal and before dental implantation, patients underwent morphometry of the bone tissue using a surgical caliper and cone beam computed tomography (CBCT) for a comparative assessment of the morphometric parameters of the alveolar ridge. The height of the lingual/palatal wall and the buccal/vestibular wall of the socket, as well as the width at the apex of the alveolar ridge and the socket in the central part were measured.

On the seventh day after removal, a clinical assessment of the soft tissue’s regeneration was performed, based on the Watchel Early Wound Healing Index (EHI).

During dental implantation, a trephine biopsy specimen was taken for its further morphological evaluation.

Measurements of the level of stability of the dental implant were recorded using the resonance frequency analysis method with the Osstell ISQ device (Osstell, Göteborg, Sweden) at the stages of the implant’s placement and before making prosthetics.

Subsequently, the patient was transferred to the prosthodontic department for treatment, and 6 months after the completion of prosthodontic rehabilitation, an orthopantomogram was performed to assess the bone tissue in the area of the dental implants.

### 2.5. Treatment Protocol

Under local anesthesia with “Ultracaini D-S” (Aventis, Frankfurt am Main, Germany), tooth extraction was carried out by a standard protocol. Thorough socket curettage of the extracted tooth was performed. At this stage, all pathologically altered tissues were removed. It was preferred to conduct the extraction without fragmenting the tooth for maintaining the maximum volume of the resulting graft. Mechanical processing of the extracted tooth was carried out with a curette, a molt elevator, a dental excavator and a drill with diamond burs or separating disks (Figure 1).

For softening the soft tissue components located on surface of the root or coronal part (plaque, periodontal fibers), it is recommended to place the treated tooth in a 3% hydrogen peroxide solution for 1–2 min before and after mechanical processing. The procedure of mechanical processing of the tooth’s surface involved, in particular, the removal of the affected tooth enamel, dentin or cementum with visible changes.

After mechanical processing, the surface of the tooth was dried and placed in the dental mill (Figure 2).

For convenience during forming and subsequent complete filling of the socket by the autologous graft, the processed tooth was crushed in the mill for 3–5 s. The specified time was sufficient to obtain particles with a size of 300–1000 µm. As a result, a finely dispersed substance was formed, consisting predominantly of dentin. With the use of the dental mill, vibration sorting (20 s) of the formed particles by size was achieved.

Particles of lesser size (less 300 µm) were weeded out with the use of built-in sieves and were not further used for preservation. Experimental studies have noted a significant increase in the level of resorption with the use of “small” particles because of the active impact of phagocytes on them in the regeneration zone [24].

In a glass container, the resulting graft was processed for 15 min using a solution containing 0.5 M caustic soda (NaOH) and 30% alcohol, resulting the material being freed from organic fragments, bacteria and toxins (Figure 3).

After processing the particles were dried with sterile gauze. The pH of the material was then neutralized with a sodium phosphate buffer (physiological) solution for 5 min.

Next, the material was once again dried using sterile gauze. After antiseptic processing, the received material was placed in the socket, which was previously curetted (Figure 4). The edges of the wound were sutured using a synthetic braided suture material; tight suturing was unnecessary.

### 2.6. Postoperative Care

By the end of the tooth extraction, the patient applied cold to postoperative area for 15–20 min, and a standard course of antibacterial and anti-inflammatory therapy was prescribed. The patient was not allowed to brush her/his teeth for the next 24 h. Oral baths with a 0.05% “Chlorhexidine” or “Miramistin” were assigned.

### 2.7. Statistical Analysis

Statistical analysis of the study’s results was performed using Statistica 12.0 software. According to the results of the test for the normality of distribution, to assess the significance of differences in the dependent and independent groups, the Student, Wilcoxon, Kruskal–Wallis and Mann–Whitney tests were implemented using Bonferroni correction.

## 3. Results

### 3.1. Evaluation of Soft Tissue Healing after Preservation

The study involved 80 patients of both sexes, of whom 46 were women (57.5%) and 34 were men (42.5%). The average age of females was 45 years; that of males was 46 years.

No cases of alveolitis or inflammation in the removal area in the postoperative period were found in any clinical case (Table 1).

This, in our opinion, was due to the fact that the use of osteoplastic material stabilized the blood clot in the socket, which had a positive effect on the regenerative processes. The best results for soft tissue healing were achieved with PRGF. Groups 1, 3 and 4 showed relatively similar results in terms of the soft tissue’s recovery after tooth extraction. No complications were recorded in these groups. Clinical analysis and the photo protocol 14 days after extraction showed that the socket of the extracted tooth in all groups was completely covered with newly formed soft tissues (Figure 5).

### 3.2. Evaluation of the Healing of Bone Tissue after Conservation

Before tooth extraction and before installation of a dental implant, the morphometric parameters of the alveolar ridge were measured (Table 2). In the PRGF group, there was a significant level of bone resorption across both the width and height of the socket. In the remaining groups, the results of bone tissue formation were similar. There were no statistically significant differences among the four groups, distributed depending on the preservation material and the width of the base (*p* = 0.36).

In Group 2, using PRGF as a preservative material, the level of resorption was the highest. The vertical loss of bone tissue of the vestibular wall of the socket averaged 2.2 mm (24.2%), and the size of the lingual-palatal wall decreased by 1.9 mm (21.3%). In the horizontal direction, the level of resorption of the socket in the region of the apex of the alveolar process was 1.8 mm, which was 23.3% of the initial values.

The results of CBCT were consistent with the findings obtained from the morphometry of the alveolar ridge. The minimum level of reduction was detected when the xenograft and ADM were placed into the socket (Table 3).

Analysis of the presented data showed that there was a statistically significant (*p* < 0.001) relatively uniform decrease in the parameters, regardless of the groups in the comparison. The median value of the vertical loss of the vestibular wall of the socket in the xenograft group was at the level of 4.4% of the initial value (Figure 4). Similar indicators were noted in the group with the use of an autologous dentin matrix: 5.2%. A significantly higher (*p* < 0.05) level of resorption in the vertical direction was seen in the group using the material based on hydroxyapatite (16.8%), and the highest was when PRGF was used (21.9%).

When analyzing the resorption of bone tissue in terms of width, the best results were demonstrated in Groups 1 and 3. In the group using xenografts, the level of horizontal resorption was 3.7% of the initial value. When using ADM, bone loss also turned out to be insignificant (3.7%), and when hydroxyapatite was used, it was 15.8%.

### 3.3. Evaluation of the Morphological Picture of Bone Trephine Biopsy Specimens

The next step was a comparative assessment of the morphological picture of the bone that regenerated 4 months after the preservation of the tooth socket.

In Group 1, where the xenograft was used as a preservation material, the histological results showed that the material particles were easily distinguished from the newly formed bone; a significant amount of new bone formed among the granules. A large amount of connective tissue was found in the obtained regenerated bone. Close contact between the newly formed bone and a small part of the xenograft particles was visualized (Figure 6a).

In the study of bone regeneration achieved after the preservation of sockets using plasma rich in growth factors (PRGF), the outer layer of the buccal alveolar bone showed corticalization and the formation of lamellar tissue, which indicated the positive effect of plasma on the regeneration of bone tissue. A significant amount of residual platelets was also found in the regenerated zone, as well as a significant amount of connective tissue in the regenerated zone (Figure 6b).

In Group 3, where ADM was used as a preservation material, the regeneration from the socket right after the preservation with autologous dentine fragments was represented by bone structures of varying degrees of severity and maturity. After 14 weeks, the socket area was filled with cancellous bone, in which well-developed bone trabeculae anastomosing with each other were noted. At the periphery, they were represented predominantly by reticulo-fibrous bone tissue, while zones of lamellar structure were noted closer to the base. There were synthetically active osteoblasts on the surface of the beams (Figure 6c).

A morphological assessment of the group using hydroxyapatite as a preservation material, revealed trabeculae from the newly formed bone in the resulting regenerated bone tissue, penetrating small fragments of biomaterial inside the trabeculae, with intense biodegradation and without an inflammatory infiltrate. Granules of bone material were surrounded by regenerated bone and fibrous connective tissue, as well as significant areas of coarse fibrous bone tissue (Figure 6d).

Analysis of the obtained trephine biopsy specimens from the preservation zone showed that a high level of osteoregeneration was achieved in the groups where ADM and hydroxyapatite were used as preservation materials due to the formation of a significant amount of coarse fibrous and lamellar bone tissue (Table 4).

On the basis of the morphometric data, it must be concluded that ADM has pronounced osteoinductive properties. In particular, this was supported by a larger volume of “young” coarse fibrous bone tissue (42.0%), which formed after its implantation in the tooth socket. A significant part of the “young” bone tissue in this case should be considered as a prognostically favorable springboard for further osteogenic differentiation and its modeling into lamellar bone tissue with the necessary strength properties. When using hydroxyapatite, this process occurred somewhat more intensively, where the area of coarse fibrous bone tissue was 38.3% and that of lamellar tissue was 34.3%. In the groups with the use of ADM and hydroxyapatite, the total area of bone tissue in the biopsy specimens was 62.6% and 72.6%, respectively, which were the highest results among the compared groups. Somewhat lower results were determined in the groups using xenografts and PRGF (36.7% and 30.2%).

An important fact is that both in the ADM group and in the hydroxyapatite group, there were, actually, no areas occupied by the excess of connective tissue (no more than 25%). When a xenograft was used, this indicator increased somewhat and amounted to 39.4%. When the socket was preserved with PRGF, the area of connective tissue in the obtained biopsy specimens was 69.7%.

The formation of such tissue is usually considered as an unfavorable outcome, since, due to the high probability of fibrosis, conditions are created in the bone wound that prevent the growth of bone tissue. In fact, the area of osteoregeneration is scarred. In addition, such an outcome for a period of 4 months from general morphological and general pathological processes can be considered as a consequence of the encapsulation of osteoplastic material as a result of inflammation; in other words, in these cases, the granules of the material act as a foreign body for tissues and induce not histotypic osteogenic regeneration, but inflammation with further outcomes in bradytrophic scars.

### 3.4. Evaluation of the Stability of Dental Implants

In accordance with the program and plan of the study, 151 dental implants were installed in patients, and at the final stage, the effectiveness of dental implantation after the preservation of the socket of the extracted tooth using various osteoplastic materials was evaluated. The postoperative period was uneventful, without inflammation.

Analysis of the level of stability showed that at the stage of the implants’ installation, this indicator in the compared groups fluctuated in the range of 52.4–58.0 ISQ (Table 5). Thus, based on the recorded average statistical parameters, adequate primary stability was achieved in all clinical groups. There were no statistically significant differences between the patient groups of with xenograft and ADM and between those with xenografts and hydroxyapatite (*p* = 0.22, *p* = 0.38).

At the stage of prosthodontic rehabilitation, statistically significant differences occurred between the ISQ values of the groups with xenografts, ADM and hydroxyapatite on the one hand, and PRGF on the other hand (in all three pairs of comparisons, *p* < 0.001). Whereas between the values in the xenograft and hydroxyapatite groups, the differences were not statistically significant (*p* > 0.05).

In assessing the dynamics of changes in each individual group, it must be stated that the value of the stability index of dental implants increased in the groups with xenografts, ADM and hydroxyapatite approximately equally (by 25.8%, 23.5% and 23.4%, respectively). In all these cases, the differences between the ISQ value at the stages of the implants’ placement and prosthodontic rehabilitation were statistically significant (*p* < 0.001).

### 3.5. Remote Evaluation of Installed Implants

During the follow-up examination of patients 6 months after fixation of the prosthetics, the effectiveness of the treatment was continued to be evaluated on the basis of clinical data and data from X-ray examination methods (OPG).

For evaluation, a number of criteria were used, such as the absence of patient complaints considering the area of the dental implants and prosthetics, the absence of hyperemia of the mucous membrane in the area of the implants, the absence of pain in the implantation zone and the immobility of prosthetic structures (Table 6).

All patients underwent a control X-ray examination in the form of an OPG, which determined the absence of a zone of continuous reduction in density around the implants. At the same time, the level of vertical resorption of bone tissue in the cervical area was allowed to be no more than 0.1 mm. The efficiency of implantation, regardless of the method of preimplantation preparation, was 100%.

## 4. Discussion

The extraction of a tooth leads to bone loss, which causes resorption of the alveolar process. Due to resorption of the alveolar bone, soft tissue shrinkage occurs [11,12]. Even the most conservative tooth extraction can cause bone resorption and lead to the need for a bone augmentation procedure when placing an implant, especially in the aesthetic area [26]. A large loss of buccal cortical bone occurs in patients with a thin periodontium [27]. In the presence of marginal bone pathology or traumatic tooth extraction with the subsequent loss of a bone wall, fibrous tissue will grow into part of the extraction socket and prevent normal bone healing and regeneration [28].

To date, many scientific works have demonstrated a decrease in the degree of bone tissue resorption when performing preservation of the sockets of extracted teeth in the preimplantation period [29,30]. The use of this method also reduces the risk of complications and the cost of treatment. A very important point is the quality of the regenerated bone tissue to achieve satisfactory stability of the dental implant [31].

In most studies that have examined the quality of newly formed bone tissue after preservation, it has been argued that the ideal bone graft material should not only have osteoconductive properties but should also promote osteoinduction and osteogenesis. Only autologous bone has these three properties, and it is still considered the gold standard for bone augmentation procedures [9,11,20]. However, the increased scale and time of surgery, the limited availability of autologous bone and postoperative discomfort can lead to the use of alternative bone substitutes for bone regeneration [32]. The majority of authors came to such conclusions, considering that the frequency of complications in the sampling of autologous bone grafts is from 8.6% to 20.6% [33]. These shortcomings, as well as the fact that medicine is developing towards minimally invasive methods of treatment, have led to the search for alternative materials for bone tissue regeneration.

One such material is the platelet concentrate PRGF, which is widely used in dental and maxillofacial surgery to stimulate tissue regeneration and prevent postoperative discomfort [34]. Many studies have claimed that the use of PRGF has a positive effect on the regeneration of soft and hard tissues [32,35].

In the last decade, xenografts have been actively used for alveolar ridge augmentation due to their osteoconductive properties. To obtain xenogenic materials, products of animal origin are used [36,37]. According to the WHO’s requirements, biomaterials obtained from cattle must be free from prions and other proteins if they are exposed to temperatures exceeding 800 °C [38,39,40]. However, the authors believe that complete resorption and replacement of the material takes about 12–20 months [41]. In addition, for a number of reasons, one of which is the lack of osteoinductive properties, the use of xenograft in dentistry remains limited and does not allow its use in everyday practice [42].

Another full-fledged clinical product for guided bone regeneration and alveolar ridge preservation is material of human origin (allografts) [43]. Allografts are available in several variants: corticospongiosus, cancellous blocks and granules [44]. Bone for the production of this material is collected from living donors who have undergone femoral head replacement surgery, and undergoes comprehensive testing and purification in accordance with regulations [45]. Despite the advantages of allogeneic bone tissue, its widespread use is limited by difficulties in the material’s procurement, transportation, preservation and sterilization, including those due to partial loss of the osteoinductive component of the resulting product. In addition, when using allogeneic materials, the possibility of an immune conflict and transmission of a number of diseases cannot be ruled out.

Recent systematic reviews by Avila-Ortiz et al. (2019) and Troiano et al. (2018) confirmed the results of the study, indicating that bone-volume-sparing procedures effectively reduced the rate of loss of vertical ridge height by 1.65–1.72 mm compared with unassisted healing and preservation techniques [46,47,48]. However, according to a systematic review, there is some evidence that indicates that resorption cannot be avoided even after socket preservation has been performed [32]. The results of our study are consistent with these findings, since in all experimental groups, resorption was observed in the width and height of the alveolar ridge even after preservation. Thus, the procedure of extraction with subsequent socket preservation does not result in complete dimensional stability, but is intended to reduce the loss of ridge size compared with the areas left to heal naturally after tooth extraction. This approach is supported by other studies, which reported that preservation with any material is superior to spontaneous healing, and the use of different scaffold materials may help reduce resorption of the socket volume after extraction [11,12,14,17,27,30].

Bone graft materials are selected based on their ability to serve as a scaffold, maintain space for regeneration of new bone and have only osteoconductive activity. ADM has osteoinductive properties. It serves not only as a scaffold for the regeneration of new bone, but also stimulates the differentiation of mesenchymal cells into osteoblasts [15,16,17,18,19].

The results of the morphometric study showed that the tissues’ reaction in the regeneration zone is quite variable; however, the best results were obtained in the groups with ADM, xenografts and hydroxyapatite. However, there are a number of publications that indicate the need for a barrier membrane during socket preservation, which can provide more predictable bone formation at the apex of the alveolar ridge [31,49].

The study showed that the preservation of the socket after tooth extraction in the preimplantation period led to a significant decrease in the level of bone resorption, prevented the development of alveolitis and ensured the installation of dental implants in optimal clinical conditions without additional surgical interventions aimed at increasing the volume of bone tissue. The best results for socket preservation were observed in patients in groups using xenografts and ADM. In these groups, no postoperative complications were detected in the patients, and the minimum level of resorption was recorded, which was confirmed by the data of the morphometric analysis and CBCT. The high level of osteoregenerative properties in these materials was also confirmed by the data of the morphological picture and the resonance frequency analysis of dental implants. Further development is the expansion of possibilities for the use of ADM (guided bone regeneration, sinus lifts). It is also important to analyze the various techniques for processing the ground tooth, including an analysis of the crystal lattice of the resulting granules.

## 5. Conclusions

The obtained results indicate that the implementation of preservation measures is the most important element of preimplantation preparation, which makes it possible to avoid additional surgical interventions to eliminate bone resorption. When preserving the socket, the use of Cerabone, ADM and Kollapan-L materials is justified, and the introduction of PRGF is recommended with the use of osteoprotective materials.

## Figures and Tables

**Figure 1 medicina-60-01280-f001:**
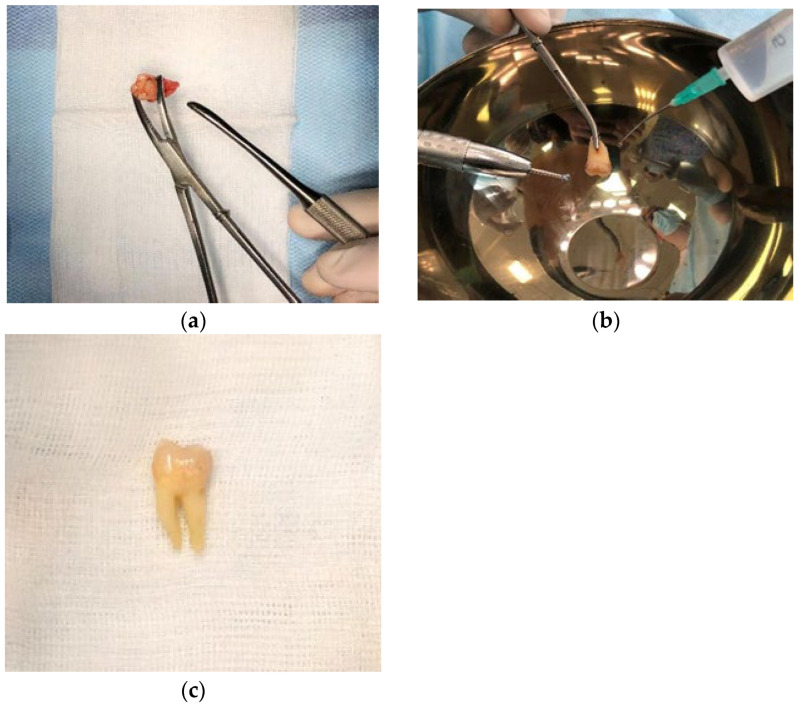
Photo of the extracted tooth before processing (**a**), at the processing stage (**b**), and processed and prepared for grinding (**c**).

**Figure 2 medicina-60-01280-f002:**
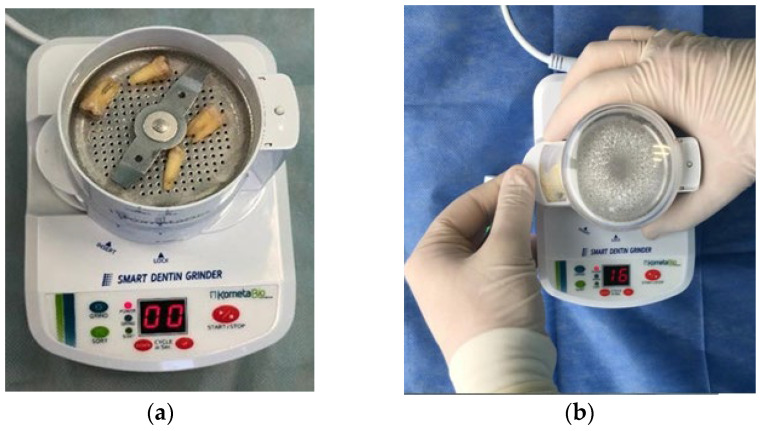
Photo of extracted teeth before grinding with a tooth mill (**a**) and after (**b**).

**Figure 3 medicina-60-01280-f003:**
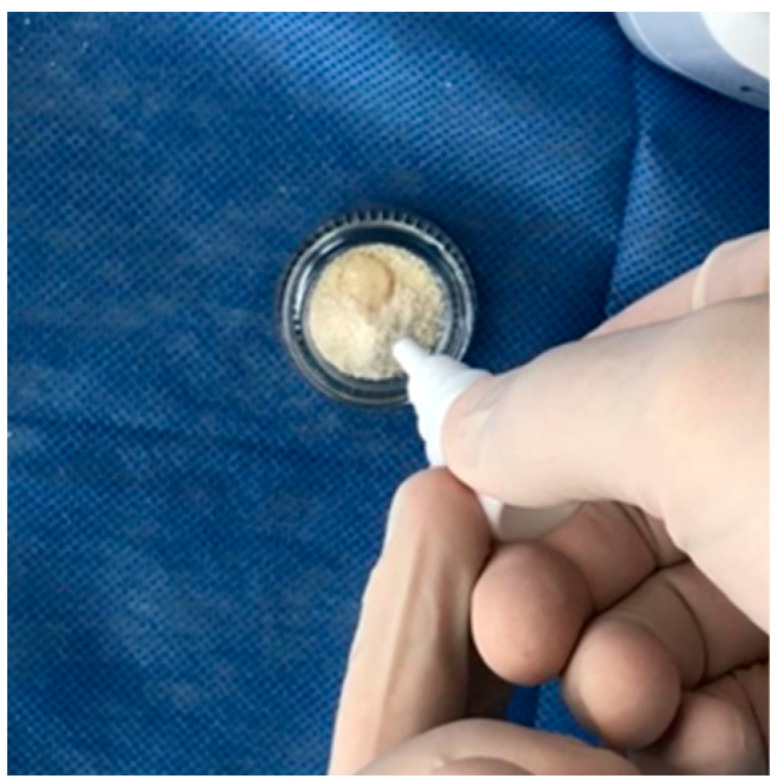
Photo of the antiseptic processing of the received particles.

**Figure 4 medicina-60-01280-f004:**
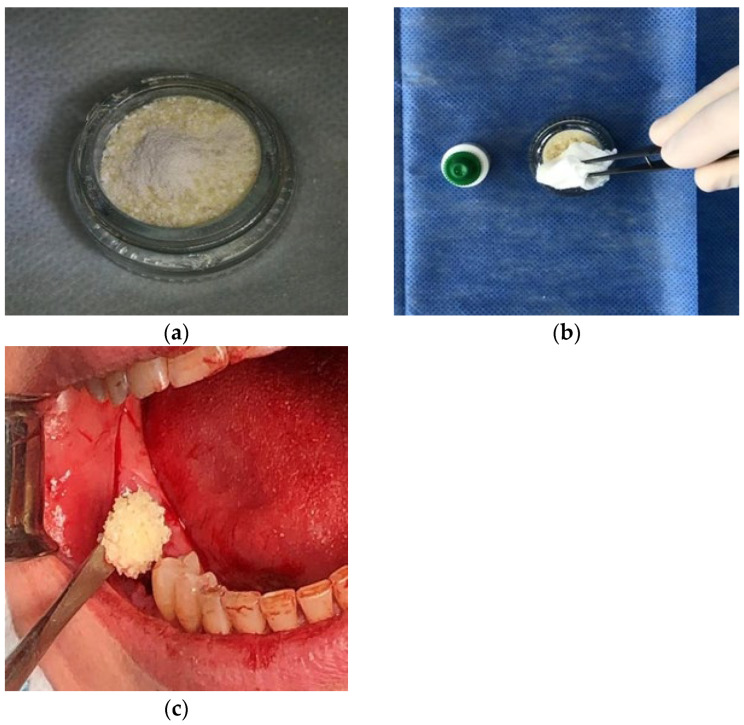
Photo of the processed crushed autologous dentin matrix (**a**), the drying stages (**b**) and the placement of the material into the socket (**c**).

**Figure 5 medicina-60-01280-f005:**
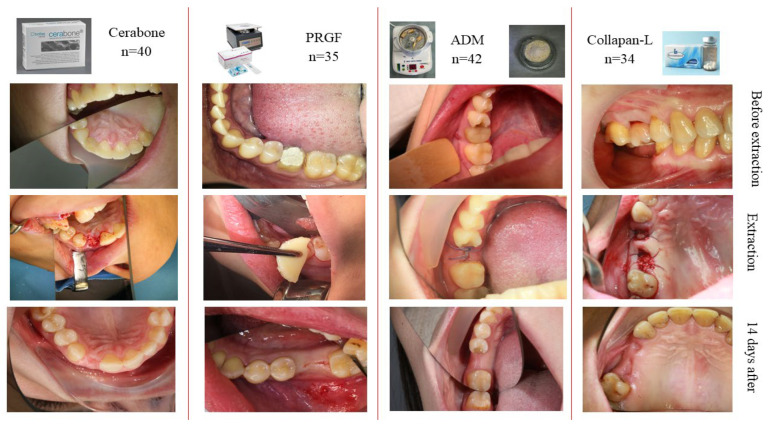
Intraoral photographs before tooth extraction (**top row**), at the extraction stage (**middle row**) and on the 14th day after the preservation of the socket (**bottom row**).

**Figure 6 medicina-60-01280-f006:**
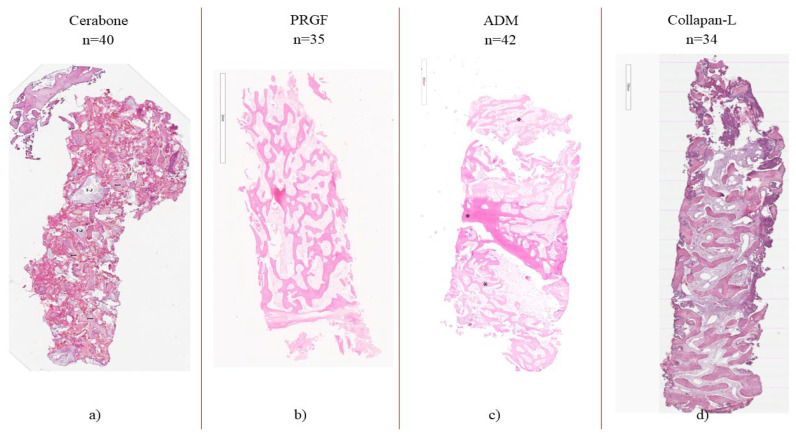
(**a**) Regeneration from a socket filled with xenograft granules, 16 weeks (female, 38 years old): 1, bone regenerated with polygonal bone beams; 2, granules of xenograft. (**b**) Fragments of the regeneration from the socket filled with plasma rich in growth factors, PRGF, 16 weeks (female, 32 years old); (**c**) regeneration from a socket filled with fragments of the autologous dentin matrix, 14 weeks (female, 42 years old). * Trabeculae of newly formed; in the intertrabecular space, reactively altered connective tissue (granulation tissue) is seen. (**d**) Regeneration from a socket filled with hydroxyapatite granules at 4 months (female, 48 years old): regenerated bone of a cancellous structure; in the intertrabecular space, there is fibrous connective tissue. All biopsies are displayed at 40× magnification. Staining: hematoxylin and eosin.

**Table 1 medicina-60-01280-t001:** Evaluation of the healing of the socket of the extracted tooth with one-stage preservation on the seventh day after surgery using the Watchel Early Wound Healing Index (EHI).

The Early Wound Healing Index (EHI)	Xenograft *n* = 40	PRGF *n* = 35	ADM *n* = 42	HaP *n* = 34
EHI 1: Complete flap closure without a fibrin line in the interproximal area	24 (60.0%)	28 (80.0%)	25 (59.5%)	19 (55.9%)
EHI 2: Complete flap closure with a fine fibrin line in the interproximal area	14 (35.0%)	6 (17.2%)	16 (38.1%)	11 (32.4%)
EHI 3: Complete flap closure with fibrin cloth in the interproximal area	2 (5.0%)	1 (2.8%)	1 (2.4%)	3 (8.8%)
EHI 4: Incomplete flap closure with partial necrosis of the interproximal tissue	-	-	-	1 (2.9%)
EHI 5: Incomplete flap closure with complete necrosis of the interproximal tissue	-	-	-	-

(*n* = number).

**Table 2 medicina-60-01280-t002:** Table of the comparative analysis of the median values of the indicators before removal (T1) and in the preimplantation period (T2) according to morphometry.

Time of Measurement	Xenograft *n* = 40	PRGF *n* = 35	ADM *n* = 42	HaP *n* = 34
T1				
Mean buccal-vestibular height (mm), SD	8.4 [8.2; 9.1]	9.1 [8.7; 9.7]	9.2 [9.0; 9.3]	9.6 [9.3; 10.2]
Mean palatal-lingual height (mm), SD	8.2 [8.0; 8.5]	8.9 ± 0.8	8.9 [8.5; 9.1]	9.5 ± 0.7
Mean width of the base of the alveolar ridge (mm), SD	7.6 ± 0.4	7.6 ± 0.3	7.9 ± 0.4	7.3 ± 0.9
Mean width of the top of the alveolar ridge (mm), SD	7.7 ± 0.3	7.7 [7.6; 8.0]	7.7 ± 0.4	7.9 ± 1.2
T2				
Mean buccal-vestibular height (mm), SD	8.1 [7.9; 8.7]	6.9 [6.8; 7.5]	8.65 [8.55; 8.84]	7.8 [7.6; 8.5]
Mean palatal-lingual height (mm), SD	8.0 [7.8; 8.2]	6.8 [6.7; 7.0]	8.5 [7.9; 8.7]	7.9 ± 0.6
Mean width of the base of the alveolar ridge (mm), SD	7.4 ± 0.4	5.89 [5.79; 5.93]	7.5 ± 0.4	6.0 ± 0.7
Mean width of the top of the alveolar ridge (mm), SD	7.5 [7.3; 7.6]	5.93 [5.89; 5.99]	7.3 ± 0.3	6.9 [5.7; 7.5]

*n*, number; SD, standard deviation.

**Table 3 medicina-60-01280-t003:** The results of measuring the parameters of the alveolar ridge before removal (T1) and before implantation (T2) according to CBCT.

Time of Measurement	Xenograft *n* = 40	PRGF *n* = 35	ADM *n* = 42	HaP *n* = 34
T1				
Mean buccal-vestibular height (mm), SD	9.05 [8.8; 9.7]	9.6 [8.8; 9.97]	9.6 [9.4; 9.9]	10.1 [9.7; 10.6]
Mean palatal-lingual height (mm), SD	8.8 [8.6; 9.1]	9.2 [8.6; 9.5]	9.3 [8.8; 9.6]	10.0 ± 0.7
Mean width of the base of the alveolar ridge (mm), SD	8.2 ± 0.4	8.1 ± 0.3	8.2 ± 0.5	7.6 ± 0.95
Mean width of the top of the alveolar ridge (mm), SD	8.3 ± 0.4	8.1 [8.0; 8.4]	8.1 ± 0.4	8.3 ± 1.3
T2				
Mean buccal-vestibular height (mm), SD	8.63 [8.4; 9.3]	7.3 [7.1; 7.9]	9.1 [9.0; 9.2]	8.3 [8.0; 9.0]
Mean palatal-lingual height (mm), SD	8.5 [8.3; 8.7]	7.1 [6.9; 7.3]	8.9 [8.3; 9.1]	8.3 ± 0.6
Mean width of the base of the alveolar ridge (mm), SD	7.9 ± 0.4	6.2 [6.1; 6.24]	7.9 ± 0.4	6.4 ± 0.8
Mean width of the top of the alveolar ridge (mm), SD	7.97 ± 0.3	6.24 [6.2; 6.3]	7.7 ± 0.4	7.0 ± 0.6

*n*, number; SD, standard deviation.

**Table 4 medicina-60-01280-t004:** Morphometric parameters of the state of bone regenerated 16 weeks after implantation of bone substitute materials.

Materials	Xenograft *n* = 40	PRGF *n* = 35	ADM *n* = 42	HaP *n* = 34
Area of coarse fibrous bone tissue (%)	27.1 ± 2.6	19.1 ± 1.8	39.46 ± 3.5	23.8 ± 2.9
Area of lamellar bone tissue (%)	9.6 ± 3.4	11.2 ± 2.2	69.7 ± 3.7	0
Area of connective tissue (%)	39.46 ± 3.5	20.6 ± 1.7	23.2 ± 3.9	14.2 ± 1.2
Area occupied by the material (%)	23.8 ± 2.9	34.3 ± 2.4	21.3 ± 1.94	6.0 ± 0.7

**Table 5 medicina-60-01280-t005:** Level of stability of the installed dental implants according to RFA.

Stages of Assessing the Stability of Dental Implants	Xenograft *n* = 40	PRGF *n* = 35	ADM *n* = 42	HaP *n* = 34
During the installation phase (ISQ), SD	57.4 ± 2.9	52.4 ± 2.0	56.7 ± 1.9	58 [57; 59]
At the stage of prosthetics (ISQ), SD	72.2 ± 2.6	63 [62; 65]	70 [69; 72]	71.6 ± 2.2
*p*-Value	*p* < 0.001	*p* > 0.001	*p* < 0.001	*p* < 0.001

**Table 6 medicina-60-01280-t006:** The results of assessing the success of treatment using dental implants 6 months after fixation of the prosthetics.

Criteria for Evaluation	Xenograft *n* = 40	PRGF *n* = 35	ADM *n* = 42	HaP *n* = 34
The dental implant is stable (%)	100%	100%	100%	100%
Absence of peri-implantation changes according to OPTG (%)	100%	100%	100%	100%
The amount of vertical bone loss (mm)	0.04 ± 0.03	0.06 ± 0.03	0.02 ± 0.04	0.07 ± 0.02
No pain in the implant area	100%	98%	100%	100%
Absence of mucositis	100%	100%	100%	97%

## Data Availability

No new data were created or analyzed in this study. Data sharing is not applicable to this article.

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
