# Peer review of "Comparative Effectiveness of an Autologous Dentin Matrix for Alveolar Ridge Preservation"

_medicina, 2024, doi:10.3390/medicina60081280_

Round 1

Reviewer 1 Report

Comments and Suggestions for Authors

The paper is very interesting and genuine. The authors, however, are required to improve the following points of the manuscript: 

- The abstract is well-structured, focused and concise. 

- Kindly cite this reference (DOI: https://doi.org/10.14295/bds.2021.v24i2.2364) to supplement the citations on effect of tooth loss in anterior region.

- Introduction section is short. Please add stronger statements to reflect the outstanding research question and importance of this study despite the fact that many similar studies have been published already.

- Null hypothesis/hypotheses should be added at the end of introduction section. 

- Line 83: Why did authors chose patients with chronic periodontitis specifically? you may just say following proposed inclusion criteria.

- Line 85: the word "orthopedic" is not fitting in the text.

- Line 103: the term "orthopedic rehabilitation" please follow the latest version of the glossary of prosthodontic terms.

- Description of study methodology is thorough, concise and supplemented with good figures. Although clinical pictures in some figures should be enlarged for better quality and visibility. 

- Why did authors chose to avoid allograft bone material inclusion into this study?

- Study limitations and directions for future research should be added to the discussion section. 

- Conclusion section could be expanded more to cover the study outcomes and may be listed in bullet points for clarity.

Author Response

Dear Editor and Reviewers,

We are grateful for the Reviewers’ valuable comments concerning the manuscript, “Comparative effectiveness of autologous dentin matrix for alveolar ridge preservation”. In the light of the questions and recommendations received we revised the manuscript extensively. Please find below our responses to the Reviewers’ comments one by one. We feel that the manuscript has been significantly improved and hope that the corrections meet the requirements for publication.

With kind regards,

Nikolai Redko

Q-1: The paper is very interesting and genuine. The authors, however, are required to improve the following points of the manuscript The abstract is well-structured, focused and concise.

A-1: Thank you for such a comment

Q-2: Kindly cite this reference (DOI: https://doi.org/10.14295/bds.2021.v24i2.2364) to supplement the citations on effect of tooth loss in anterior region.

A-2: Thank you, we agree with this comment, so we have cited this article [line 35]

Q-3: Introduction section is short. Please add stronger statements to reflect the outstanding research question and importance of this study despite the fact that many similar studies have been published already

A-3: Thanks for the comment. The introductory section has been added

Considering its biological characteristics, lack of allergic reactions and accessibility to the clinician and patient, autologous dentin matrix can be considered as a possible option as a graft for graft preservation in oral surgery. Despite the considerable amount of research on this problem, the improvement of pre-implantation preparation of patients with adentia after tooth extraction continues to be an extremely urgent task. The solution to this problem requires a comprehensive approach based on a unified assessment of the success of dental implantation in the post-extraction zone. The aim of our study is to conduct a comprehensive comparative analysis of osteoplastic materials after preservation of the extraction wells of extracted teeth.

Q-4: Null hypothesis/hypotheses should be added at the end of introduction section.

A-4: The null hypothesis was added [line 73-74].

Q-5: Line 83: Why did authors chose patients with chronic periodontitis specifically? you may just say following proposed inclusion criteria.

A-5: Thanks for the comment. We have changed to "in accordance with the proposed inclusion criteria." [line 87-88]

Q-6: Line 85: the word "orthopedic" is not fitting in the text.

A-6: Thank you! The term has been changed [line 87-88]

Q-7: Line 103: the term "orthopedic rehabilitation" please follow the latest version of the glossary of prosthodontic terms

A-7: Thank you! The term has been changed [line 107]

Q-8: Description of study methodology is thorough, concise and supplemented with good figures. Although clinical pictures in some figures should be enlarged for better quality and visibility.

A-8: Thanks for the comment! The clinical pictures have been magnified

Q-9: Why did authors chose to avoid allograft bone material inclusion into this study?

A-9: Dear reviewer. Comparisons with allograft are presented in the following articles. We have contributed data to the text. [line 411-421].

Q-10: Study limitations and directions for future research should be added to the discussion section.

A-10: Thanks for the comment! The data has been included in the discussion chapter. Further development is the expansion of possibilities for the use of ADM (guided bone regeneration, sinus lift). It is also important to analyse the various techniques for processing the ground tooth, including the analysis of the crystal lattice of the resulting granules [line 455-458]

Q-11: Conclusion section could be expanded more to cover the study outcomes and may be listed in bullet points for clarity

A-11: Thanks for the comment! The Conclusion section has been expanded [line 462-464].

Reviewer 2 Report

Comments and Suggestions for Authors

This manuscript is of some interest, but lacks sufficient content.

1. The biggest problem with this manuscript is that the control group is unclear and there are too many of them.

2.If we want to accurately analyze the effect of autologous dendin matrix (ADM), HA+ADM (experimental group) vs HA (control group) should be the focus of the study. But why do we need xenograft and PRGF as unnecessary control groups?

3.Why was Cerabone chosen instead of the standard Bio-Oss as a xenograft?

4. The manuscript is overall distracting and difficult to focus on.

Author Response

Dear Editor and Reviewers,

We are grateful for the Reviewers’ valuable comments concerning the manuscript, “Comparative effectiveness of autologous dentin matrix for alveolar ridge preservation”. In the light of the questions and recommendations received we revised the manuscript extensively. Please find below our responses to the Reviewers’ comments one by one. We feel that the manuscript has been significantly improved and hope that the corrections meet the requirements for publication.

With kind regards,

Nikolai Redko

Q-1: The biggest problem with this manuscript is that the control group is unclear and there are too many of them.

A-1: Dear reviewer, lines 89-98 detail which groups were involved in the study. Perhaps the most optimal control group would be a comparison with a blood clot well, however, a large number of publications have been written on this topic and bone resorption rates are extremely high with this technique

Q-2: If we want to accurately analyze the effect of autologous dendin matrix (ADM), HA+ADM (experimental group) vs HA (control group) should be the focus of the study. But why do we need xenograft and PRGF as unnecessary control groups?

A-2: Dear reviewer! Thank you for your question! This paper compares "pure, unmixed" techniques in order to obtain the most reliable results possible. There was no HA+ADM group in our study. And the comparison with xenomaterial or a preparation based on the patient's venous blood was carried out due to the widespread use of these techniques among practitioners. Also, it was important to assess the level of soft tissue healing in the hole, including in comparison with PRGF

Q-3: Why was Cerabone chosen instead of the standard Bio-Oss as a xenograft?

A-3: Thank you for your very relevant question! The choice of Cerabone xenograft was based on its prevalence and popularity in the market. Also, both of these materials belong to the xenograft group and are similar in their morphology and preparation method.

Q-4: The manuscript is overall distracting and difficult to focus on.

A-4: Dear reviewer! We have tried to fully present the material obtained during the research.

Round 2

Reviewer 2 Report

Comments and Suggestions for Authors

This manuscript has been well revised.